# Coupling Effects of pH and Dissolved Oxygen on the Corrosion Behavior and Mechanism of X80 Steel in Acidic Soil Simulated Solution

**DOI:** 10.3390/ma12193175

**Published:** 2019-09-27

**Authors:** Shuaixing Wang, Xiaole Yin, Hao Zhang, Daoxin Liu, Nan Du

**Affiliations:** 1School of Material Science and Technology, Nanchang Hangkong University, Nanchang 330063, Chinanandu1957@sina.com (N.D.); 2Institute of Corrosion and Protection, Northwestern Polytechnical University, Xi’an 710072, China; liudaox@nwpu.edu.cn

**Keywords:** X80 pipeline steel, acidic soil simulated solution, pH, dissolved oxygen, corrosion mechanism

## Abstract

In an acidic red soil environment, the corrosion mechanism of X80 steel may be closely related to the pH value and oxygen content, but it has not yet formed a systematic understanding. In this paper, the coupling effects of pH and dissolved oxygen on the corrosion behavior and mechanism of X80 steel in an acidic soil simulated solution were further analyzed by electrochemical methods and three-dimensional video microscope. Results showed that the hydrogen reduction reaction was almost the only cathode process in the anoxic and low pH system, and small and dense pits were present on the electrode surface. pH value increased, the pits decreased, but the size of pits increased. In the oxygen-adequate system, oxygen-consuming (OC) corrosion preferentially occurred, and a protective corrosion product layer (including FeOOH, Fe_3_O_4_, etc.) might be formed accordingly, but the proportion of hydrogen evolution (HE) increased and the product layer had defects at a low pH environment. The specific corrosion mechanism of X80 steel in an acidic soil simulated solution is described in the relevant models.

## 1. Introduction

The corrosion of buried pipeline steel in the soil seriously threatens the safe operation of the pipeline [1,2,3]. Currently, various factors affecting soil corrosion of pipeline steel, including temperature, pH, water content, salinity, microbial and other factors have been extensively studied [4,5,6,7,8,9,10,11], but most studies have focused on the anode process of corrosion. It is well known that the soil corrosion of pipeline steel is an electrochemical process. The conjugated cathode process is always accompanied by the anodic dissolution of the steel. O_2_, H^+^ and Fe oxides (FeOOH), etc., are considered to be the main oxidants affecting the corrosion of X80 steel [6,10,12,13,14,15]. 

Research has shown that water and oxygen are the key factors for the cathodic and anodic reactions of corrosion on the metal surface, and high oxygen concentration causes corrosion only when there is moisture present in the soil [11]. The corrosion of steel in neutral/alkaline soil belongs to aerobic corrosion, and the corrosion mechanism is closely related to oxygen content [14,15]. In a saline-alkali soil medium, the oxygen activation reaction controls the corrosion process of pipeline steel [16]. However, the corrosive properties of acid red soil are different from other types of soil. Red soil generally has high water content, low oxygen content and high resistivity (>100 Ωm) [10,17,18], and as such, the red soil should be milder. However, the long-term burial results show that the red soil is extremely corrosive, and the corrosion rate of carbon steel in Yingtan soil (a typical representative of acidic red soil in China) exceeds 0.1 mm/a [19]. Therefore, there may be other factors that have not been considered for the strong corrosiveness of anoxic red soil.

It has been proven that oxygen reduction may no longer be the dominant reaction for the corrosion cathode process of steel in anoxic soil environments [20,21]. In acidic red soil environment (pH = 3.5~5.0), the corrosion mechanism of steel may be converted into hydrogen depolarization with the consumption of oxygen. Some studies have also confirmed that the oxygen content in water-saturated acidic soils is very low, and the oxygen diffusion process becomes a control step for steel corrosion under this condition [20,22]. Our previous works [23,24] have studied the corrosion behavior of X80 steel under partial pH and dissolved oxygen (DO) conditions; meanwhile, the contribution rate of oxygen depolarization and hydrogen reduction to the corrosion of X80 steel under different conditions has been also analyzed by the homemade hydrogen collection testing, and the cathodic corrosion reaction of X80 steel under different conditions has been preliminarily analyzed [24]. However, in general, the corrosion mechanism of X80 steel in red soil environment with different conditions has not been systematically understood.

In this work, a series of acidic soil simulation environments, such as low pH/anoxic (or oxygen saturation) and high pH/anoxic (or oxygen saturation), have been established according to the composition, acidity and gas permeability of red soil in South China. On this basis, the coupling effects of pH and DO on the corrosion behavior and mechanism of X80 steel in the acidic soil simulated solution were further analyzed by the polarization curve and electrochemical impedance spectroscopy (EIS), and the relevant corrosion model was established. The research results will help to deepen the understanding of the corrosion mechanism of pipeline steel in the acid red soil system and of the corrosion control for pipeline steel.

## 2. Experimental

### 2.1. Sample Preparation

API X80 pipeline steel, with a chemical composition (wt.%) of 0.036 C, 0.197 Si, 1.771 Mn, 0.012 P, 0.184 Mo, 0.110 Nb, 0.019 Ti, 0.223 Cr, 0.278 Ni, 0.220 Cu, 0.021 Al and balanced with Fe, was used as the specimen in this work. The X80 pipeline steel plate was wire-cut into 10 mm × 10 mm × 5 mm. These samples were soldered to copper wires and then embedded in epoxy resin to act as the working electrode. All the samples were finished by wet-grinding with a series of emery papers from 500 to 1200 grit, cleaned thoroughly with alcohol and deionized water in turn and then dried in an air flow.

Yingtan (28°15/ N, 116°55/ E) is regarded as the representative region of acidic red soil in South China. The simulated solution is prepared according to the data obtained by the analysis of Yingtan soil [24,25,26]. The chemical compositions of the simulated solution are given in Table 1. The pH of the solution was adjusted to 3.0~5.5 using 1.0 M H_2_SO_4_. Besides, the oxygen content in the solution was adjusted by introducing high purity argon (purity > 99.99%) or high purity oxygen (purity > 99.5%) into the simulation solution using the aeration device shown in Figure 1a. The specific oxygen concentration in the solution was measured using a dissolved oxygen meter (STARTER 300D, Ohaus, NJ, USA) with an accuracy of 0.01 ppm. During the aeration process, we first opened the A vent and closed the B and C vents, and then passed Ar or O_2_ into the liquid storage bottle I. When the required oxygen concentration of the simulated solution in bottle I was reached, the A and B vents were exchanged. After exchanging, the B vent was closed, the C vent was opened and the simulated solution in bottle I was pressed into the II electrochemical cell by Ar/O_2_ to ensure the oxygen content of the electrochemical system. The A and C vents were closed when the desired solution content was reached in the II electrochemical cell. During the electrochemical testing, the gas was not passed to reduce the disturbance. Figure 1b gives the DO content of the acidic soil simulated solution under different aeration conditions [23]. It can be seen that the DO content in the open solution was about 4.30 ppm; DO content in the solution could be reduced to 0.25 ppm and was regarded as an anoxic state by the inflating argon gas. Besides, DO content could be raised up to 20.20 ppm by inflating oxygen gas, and here the solution could be regarded as an oxygen-saturated system.

### 2.2. Electrochemical Testing

Electrochemical testing was conducted by using an electrochemical workstation (Autolab PGSTAT 302 N, Metrohm A G, Herisau, Switzerland). A typical three-electrode electrochemical cell was used in testing, wherein the working electrode (WE) was an X80 steel sample inlaid in epoxy resin with an effective area of 1 cm^2^. The counter electrode (CE) and the reference electrode (RE) was a platinum sheet and a saturated calomel electrode (SCE), respectively. In this work, all the potentials were relative to the SCE potential. During the potentiodynamic polarization test, the potential scan range was −2.0~1.0 V (vs. SCE) and the scan rate was 1 mV/s [23,24]. The electrochemical impedance spectroscopy (EIS) were acquired at the open circuit potential (OCP) over the frequency range of 0.01 Hz~100 KHz using an AC signal amplitude of 10 mV [12,13,23]. All the tests were performed at 25 °C and repeated by three specimens to ensure the reliability of the results. After the test, the corrosion morphology of X80 steel was observed by a 3D video microscope (KH-7700, Hirox Co., Ltd., Tokyo, Japan). The phase composition of corrosion product is investigated by X-ray diffraction (XRD, D8-Advance, Bruker, Karlsruhe, Germany) with a scan range from 20° to 80° (in 2*θ*)

## 3. Results and Discussion

### 3.1. Corrosion Electrochemical Behavior of X80 Steel

Figure 2 shows the potentiodynamic polarization curves and EIS spectra of X80 steel in the acidic soil simulated solution with different pH and DO content. The polarization curves were derived from our previous research paper [24]. It can be seen that the corrosion electrochemical behavior of X80 steel in the acidic soil simulated solution was closely related to the solution pH and DO concentration.

According to Figure 2a,c,e and the reference [24], it can be found that the corrosion potential (*E*_corr_) of X80 steel showed a downward trend, and the corrosion current density (*i_c_*_orr_) decreased with the decrease of DO content at the same pH value. This indicates that oxygen participated in the electrode reaction process as a depolarizer. In addition, *i_c_*_orr_ increased with the decrease of solution pH at the same DO content, which showed that the cathode process was not only related to the reduction of oxygen, but also involved the reduction of hydrogen ions. However, the influence of hydrogen ion reduction on the cathode process was relatively small.

As shown in Figure 2a,e, the cathode polarization curves of X80 steel in the simulated solution with pH ≈ 3.0 showed the electrochemical activation control characteristics, regardless of the DO content. However, the EIS characteristics changed significantly with DO content. In the low pH and anoxic (DO ≈ 0.30 ppm) environment, there are three time constants in the EIS diagram. When the DO content rose to 1.90 ppm, the low-frequency inductive arc disappeared, and the EIS spectrum consisted of the high-frequency incomplete capacitive arc and the medium-low frequency large capacitive arc. Besides, the high-frequency capacitive arc gradually became complete as the DO content increased. In general, the low frequency region in the EIS diagram reflected the speed control step of corrosion process [13,23,27], and the high-frequency loop represented the information of the corrosion products deposited on the electrode surface [12,20,23], while the occurrence of the inductive arc was related to the pitting nucleation process [28]. Therefore, it could be confirmed that the corrosion process of X80 steel at a low pH system was controlled by the electrochemical reaction step, but the corrosion behavior of X80 steel was significantly different when the content of DO in the system was different. When the system was anoxic, X80 steel mainly underwent the pitting corrosion, and the electrode surface had almost no corrosion products, so there was a low-frequency inductive arc and a very small high-frequency capacitive arc tail in this EIS diagram, as shown in Figure 2b. The corrosion of X80 steel was intensified and its surface gradually tended to be uniformly corroded with the increase of DO content, which could be proven by the decrease of the radius for the low-frequency capacitive arc and the disappearance of the inductive arc. Meanwhile, the high concentration of oxygen also promoted the formation of corrosion products, but the corrosion product layer was still loose due to the strong HE reaction in the low pH system, as demonstrated by a slight increase in the high-frequency capacitive arc.

When the pH of solution was 4.0~4.5, all of the EIS showed two time constants regardless of the DO content, but the corrosion mechanism changed with the DO content. At a high level of DO (DO > 1.90 ppm), not only did the cathode polarization curve conform to the linear polarization characteristic (Figure 2c), but also the EIS diagram showed a double capacitive arc (Figure 2d), which indicated that the corrosion of X80 steel at these systems was controlled by the electrochemical activation process. When DO content decreased to 0.85 ppm, the cathode polarization curve appeared a weak cathodic limiting diffusive platform, and a Warburg impedance line with an inclination angle of about 45° was also present at the low-frequency region of EIS (Figure 2d), indicating that the oxygen diffusion process dominated the corrosion of X80 steel [20,23]. However, the cathode polarization curve reappeared as an activation control feature, and the EIS diagram was again converted into a dual-capacity arc when the system was almost oxygen-free (DO ≈ 0.25 ppm). Now, the corrosion of X80 steel was again controlled by the electrochemical reaction process. Since the reduction of O_2_ was negligible under this system, the cathodic reaction was only related to the reduction of H^+^ at the electrode surface. It is generally believed that the slowest step in the HE process on the surface of Fe in an acidic solution may be the electrochemical desorption of adsorbed hydrogen (H_ads_) [29,30], as shown in Equation (1).
(1)$$H^{+}+H_{\mathrm{ads}}+e\to H_{2}↑$$
That is to say, the corrosion rate of X80 steel in the anoxic solution with pH ≈ 4.5 depends on the speed of Heyrovsky reaction.

In the system with a pH of 5.5, the *i*_*c*orr_ of X80 steel decreased with the decrease of DO content, indicating that the depolarization of oxygen was the main cathodic reaction of corrosion. When oxygen was rich, the EIS diagram exhibited a high-frequency capacitive arc tail and a low-frequency capacitive anti-arc (see Figure 2f), but here, the corrosion rate of X80 steel depended on the ionization process of oxygen. Moreover, the OC corrosion process was favorable for forming the stable corrosion products [23,31], and then it inhibited the anodic dissolution process so that the anodic polarization curve had a passive feature under these conditions (see Figure 2e). The capacitive arc radius increased and gradually transformed into a Warburg-like impedance line (see Figure 2f) as the DO content decreased, indicating that the corrosion behavior of X80 steel changed with DO content. It can be considered that the diffusion rate of oxygen could not keep up with the charge transfer rate when oxygen was deficient (DO = 0.25 ~ 0.90 ppm); meanwhile, the micro-cathodes on the electrode surface would competitively adsorb a little bit of oxygen into this system and make the diffusion path of oxygen become longer. Therefore, the corrosion process of X80 steel showed a large oxygen diffusion resistance, and a limiting diffusion platform was also present in the cathodic polarization curve (see Figure 2e).

### 3.2. Corrosion Morphology and the Composition of Corrosion Products for X80 Steel

Figure 3 gives the corrosion morphology of X80 steel after EIS testing in the acidic soil simulated solution with different pH and DO content. It can be seen that there was a significant difference in the coverage of the corrosion products on the sample surface and the size and shape of corrosion holes under different corrosion conditions.

Under the system with low level of DO, the surface of X80 steel was dominated by pitting corrosion regardless of the solution pH; however, the number and size of the corrosion holes varied with the pH value. At the low pH solution (pH ≈ 3.0), the corrosion pits were small and the number was less, as shown in Figure 3a; when the pH value increased, the number of corrosion pits decreased, but the size of pits became larger, as shown in Figure 3b,c. At the same pH value, the corrosion of X80 steel was aggravated with the increase of DO content. When the DO content was 4.30 ppm, there were close and numerous corrosion pits on the sample surface, even forming pitting rings or pitting clusters. However, the sample surface had a tendency to form a corrosion product film in the oxygen-saturated system. When the solution pH was 3.0 and the oxygen was close to saturation, most areas on the sample surface had yellow corrosion products, but the product film was loose and pitting holes were still present, as shown in Figure 3c. In the oxygen-saturated system with pH of 5.5, the surface of the X80 steel was covered with a thin, uniform corrosion products layer, as seen in Figure 3i.

Figure 4 shows the XRD patterns of the corrosion products for X80 steel after EIS testing in the typical acidic soil simulated solution. It can be seen from Figure 4a that there were few corrosion products on the electrode surface in the low pH (pH ≈ 3.0)/anoxic (DO ≈ 0.25 ppm) environment, and only FeCO_3_ and Fe_3_O_4_ were detected. With the increase of pH and DO content, the number of corrosion pits on the electrode surface decreased and the corrosion products increased gradually. When the solution pH was 4.5 and the DO content was 4.30 ppm, the corrosion products, including γ-FeOOH, Fe_3_O_4_ and FeCO_3_, were sporadically distributed on the electrode surface, as shown in Figure 4b. If the DO content continued to increase, a thin and uniform corrosion product layer was formed on the surface of X80 steel. XRD analysis showed that the main components of this corrosion product layer were γ-FeOOH and Fe_3_O_4_, and a small amount of α-FeOOH and Fe_2_O_3_ were also present.

### 3.3. Corrosion Model of X80 Steel in the Acidic Soil Simulated Solution

It has been confirmed that H^+^ and O_2_ were the main corrosion depolarizers in the acidic soil simulation solution. The different ratios of the HE corrosion and OC corrosion at different pH and DO content determined the corrosion type, corrosion mechanism and corrosion morphology of the X80 steel. Based on our previous research [23,24] and the results of this work, the corrosion models of X80 steel in the acidic soil simulated solution under different conditions were established and can be seen in Figure 5.

As shown in Figure 5a, X80 steel always preferentially underwent OC corrosion in the oxygen-containing acidic soil simulated solution. The proportion of hydrogen depolarization increased with the consumption of DO, and HE volume increased. X80 steel had the smallest *i*_*c*orr_ and the lowest corrosion rate under the critical DO content at which the cathodic reaction changed.

Under the low pH (pH ≈ 3.0~3.5) and anoxic (DO ≈ 0.25 ppm) environment, H^+^ was almost always the only oxidant. The corrosion process of X80 steel was controlled by hydrogen reduction reaction; the anode and cathode reactions are shown in Equations (2) and (3), respectively [10,20,24]. Under this system, the low potential regions (such as inclusions and pits) on the electrode surface were prone to develop into pitting nucleation sites [32,33], resulting in a large number of corrosion holes. In addition, the depolarization of oxygen can be ignored in this system due to the absence of oxygen, and the Fe^2+^ produced by the anode process might react more with HCO_3_^−^ in the solution to form a little FeCO_3_ by Equation (4) [34], which is also confirmed by Figure 4a. The corrosion model under this condition is shown in Figure 5b.
Fe −2e → Fe^2+^(2)
2H^+^ + 2e → H_2_↑(3)
Fe^2+^ + HCO_3_^−^ → FeCO_3_ + H^+^(4)

When the DO content in the low pH solution (pH ≈ 3.0~3.5) was over 0.90 ppm, OC corrosion and HE corrosion coexisted, but the ratio of HE corrosion was greater than 50% [24]. As the oxygen content increased, the oxygen reduction reaction (see Equation (5)) was accelerated and the corrosion of X80 steel was aggravated. In the system with 4.30 ppm DO, a large number of corrosion pits and pitting clusters were present on the electrode surface, as shown in Figure 3b. However, the high concentration of oxygen not only accelerated the corrosion reaction, but also promoted the formation of corrosion products. The corrosion products, such as FeOOH and Fe_3_O_4_, were formed on the sample surface by a series of reactions (Equations (6)–(9)) [5,23,25], but the corrosion products layer was loose and had some defects due to the strong HE reaction. There was a large amount of yellow corrosion products on the electrode surface when the oxygen in the solution was saturated, but the corrosion holes were still present. The corrosion model under these conditions is shown in Figure 5c.
O_2_ + 4H^+^ + 4e → 2H_2_O(5)
Fe^2+^ + 2OH^−^ → Fe(OH)_2_(6)
4Fe(OH)_2_ + O_2_+ 2H_2_O → 4Fe(OH)_3_(7)
Fe(OH)_3_ – H_2_O→ FeOOH(8)
3FeOOH + H^+^ + e → Fe_3_O_4_ + 2H_2_O(9)

Under the anoxic environment of pH ≈ 4.5, the corrosion type, mechanism and morphology of X80 steel were similar to those under lower pH and anoxic conditions. However, HE overpotential increased as the pH value increased in the acidic solution, and the number of corrosion pits on the X80 steel surface decreased. Its corrosion model could still be explained by Figure 5b. However, $i_{O_{2}}/i_{\mathrm{Fe}}$ started to be greater than 50% when DO rose to 0.90 ppm [24]. OC corrosion began to occur preferentially, but the oxygen diffusion rate could not keep up with the charge transfer rate at this time, and the corrosion of X80 steel was controlled by the oxygen diffusion process, as demonstrated by the presence of Warburg impedance in the low-frequency region of the EIS diagram (Figure 3d). Here, the electrode surface still mainly underwent pitting; the corrosion model is shown in Figure 5d. If DO content continued to increase to 1.90~20.2 ppm, then the oxygen ionization step would begin to control the corrosion reaction, and EIS diagram would appear as two capacitive arcs. Under this system, the cathodic process was mainly the oxygen reduction reaction, but the corrosion model could still be illustrated by Figure 5c.

In the system with pH of 5.0~5.5, $i_{O_{2}}/i_{\mathrm{Fe}}$ was about 85%~95%, and the oxygen reduction reaction had an absolute advantage in the cathode process of corrosion [24]. However, the oxygen diffusion process was the speed control step when the DO content was 0.25~0.90 ppm, which could be proved by the Warbug impedance in the low frequency region of EIS. Here, the active sites on the electrode surface would be preferentially dissolved to form small corrosion pits, but the cathode and anode reaction products might react through the mass transfer process to form sporadic corrosion products (FeOOH, etc.) around the pits. The accumulation of corrosion products at the orifices would result in the formation of occlusion zones inside the pores. The high concentration solution inside the pits and the difference in oxygen concentration inside and outside the pits would accelerate the dissolution of metal in the pores, thereby forming deeper pits [28,35]. Under this condition, the corrosion model can be explained using Figure 5d. When DO rose to 1.90~20.2 ppm, the oxygen reduction reaction was the only cathode reaction. Fe^2+^ and OH^−^ reacted rapidly to form γ-FeOOH and Fe_3_O_4_, and some γ-FeOOH would gradually transform into α-FeOOH, so the surface of X80 steel was covered with a thin and uniform corrosion product layer. For this time, the corrosion model is shown in Figure 5e.

## 4. Conclusions

(1)In the acidic soil simulation solution, the corrosion mechanism and morphology of X80 steel largely depended on the coupling effect of the solution pH and DO content. In general, the increase of DO in the same pH system would accelerate the cathode process of corrosion while also promoting the formation of corrosion products. In the solution with the same DO content, the proportion of the HE reaction increased, and the corrosion aggravated as the pH decreased.(2)Under the anoxic (~0.25 ppm) and low pH (3.0~3.5) system, the hydrogen reduction reaction was almost always the only cathode process, and small and dense pits mainly occurred on the surface of the X80 steel. As the pH value increased, the number of pits decreased, but the size of pits increased. When the pH rose to 5.5, the oxygen reduction reaction occurred, but the oxygen diffusion process was the corrosion control step.(3)In the oxygen-adequate system, OC corrosion occurred preferentially for X80 steel, but the proportion of the HE reaction increased with the decrease of the pH value. A corrosion product layer (including FeOOH, Fe_3_O_4_, etc.) might be formed on the electrode surface accompanied by the reduction of oxygen; however, the HE reaction would cause defects in the product layer at a low pH environment.

## Figures and Tables

**Figure 1 materials-12-03175-f001:**
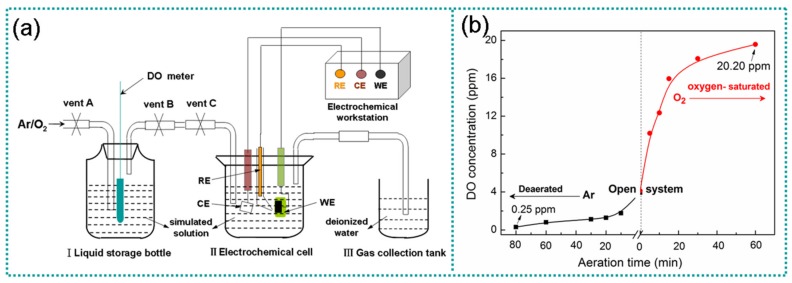
Schematic diagram of aerating device (**a**) and the oxygen content of the acidic soil simulated solution (**b**).

**Figure 2 materials-12-03175-f002:**
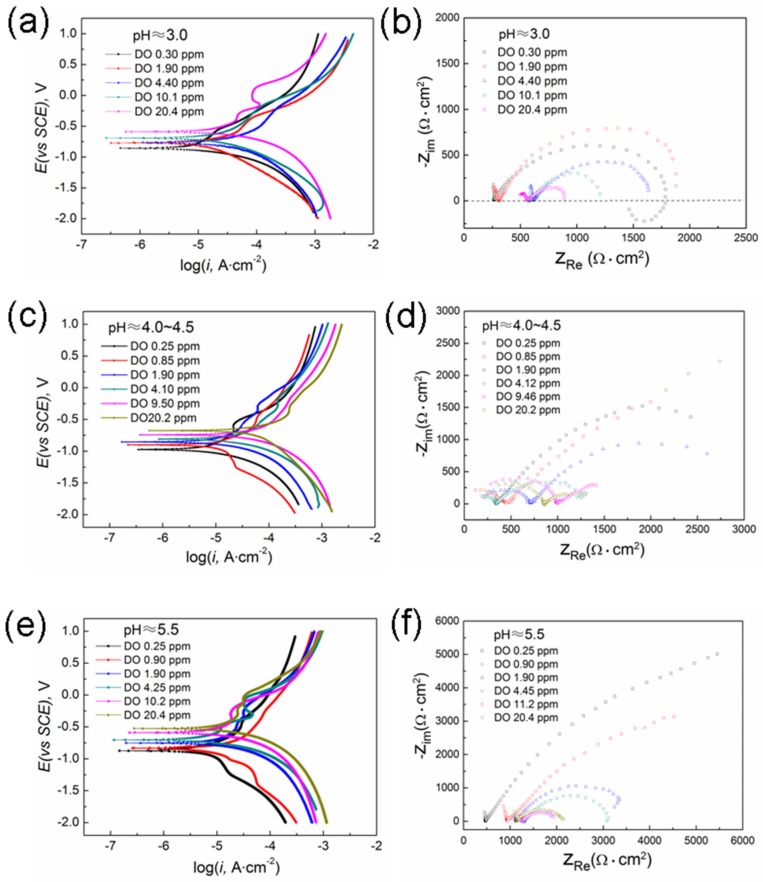
Tafel curves (**a**,**c**,**e**) [24] and Nyquist diagrams (**b**,**d**,**f**) for X80 steel in the acidic soil simulated solutions with various and pH and dissolved oxygen (DO) content.

**Figure 3 materials-12-03175-f003:**
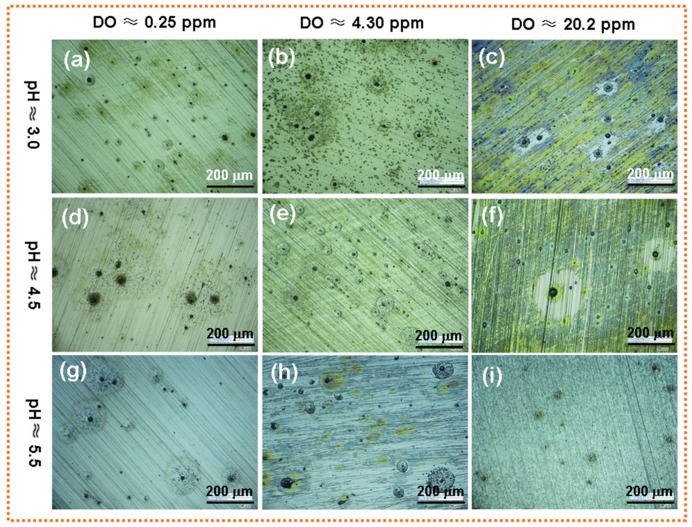
Surface OM image of X80 steel after EIS test in the acidic soil simulated solution with various pH and DO content. (**a**) pH ≈ 3.0 and DO ≈ 0.25 ppm; (**b**) pH ≈ 3.0 and DO ≈4.30 ppm; (**c**) pH ≈ 3.0 and DO ≈ 20.2 ppm; (**d**) pH ≈ 4.5 and DO ≈ 0.25 ppm; (**e**) pH ≈ 4.5 and DO ≈ 4.30 ppm; (**f**) pH ≈ 4.5 and DO ≈ 20.2 ppm;(**g**) pH ≈ 5.5 and DO ≈ 0.25 ppm; (**h**) pH ≈ 5.5 and DO ≈ 4.30 ppm; (**i**) pH ≈ 5.5 and DO≈ 20.2 ppm;

**Figure 4 materials-12-03175-f004:**
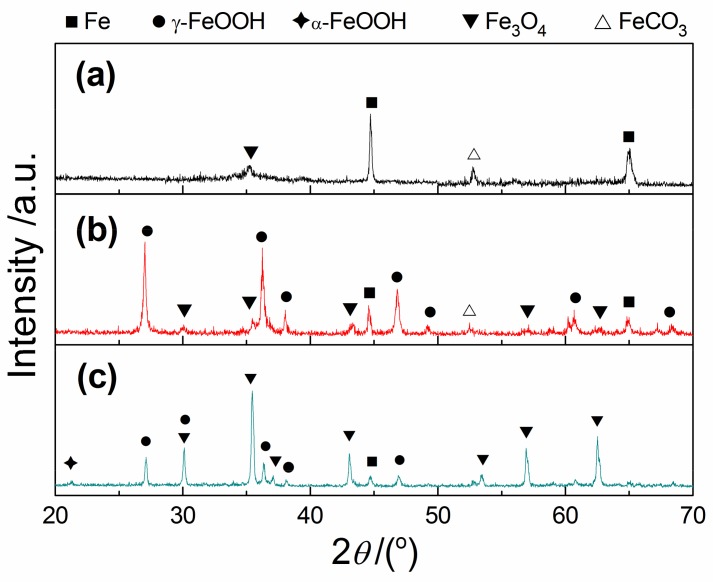
X-ray diffraction patterns of corrosion product for X80 steel after EIS test in acidic soil simulated solution under typical conditions (**a**) pH ≈ 3.0 and DO ≈ 0.25 ppm; (**b**) pH ≈ 4.5 and DO ≈ 4.30 ppm; (**c**) pH ≈ 5.5 and DO ≈ 20.2 ppm.

**Figure 5 materials-12-03175-f005:**
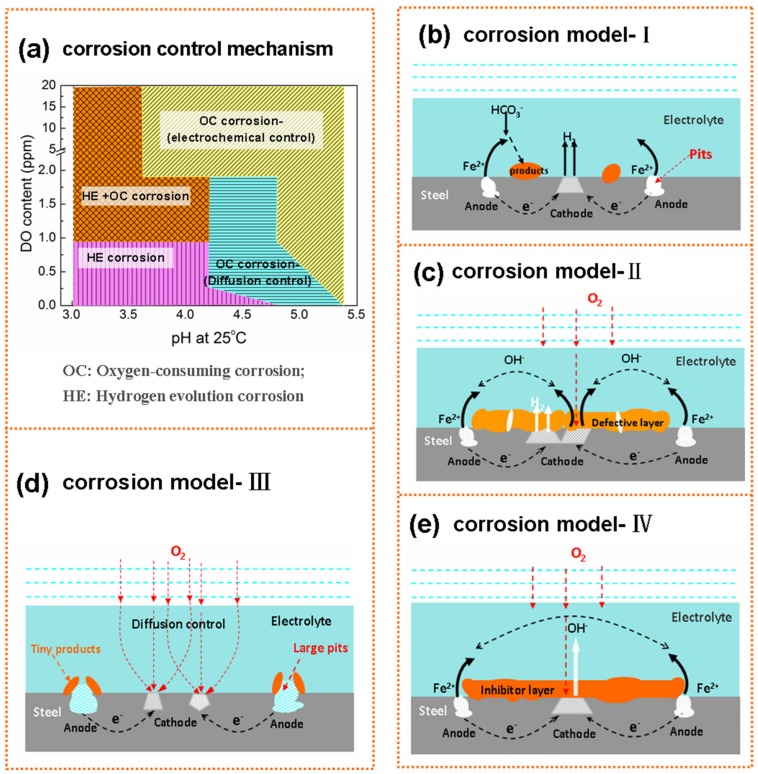
Corrosion control mechanism (**a**) and corrosion models (**b**–**e**) of X80 steel immersed in the acidic soil simulated solution with different level of pH and DO content.

**Table 1 materials-12-03175-t001:** Chemical composition of the acidic soil (Yingtan soil) simulated solution [24,25,26].

Compounds	NaCl	CaCl_2_	MgSO_4_·7H_2_O	Na_2_SO_4_	NaHCO_3_	KNO_3_
**Content (g/L)**	0.0468	0.0111	0.0197	0.0142	0.0151	0.0293

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
