# Peer review of "Coupling Effects of pH and Dissolved Oxygen on the Corrosion Behavior and Mechanism of X80 Steel in Acidic Soil Simulated Solution"

_materials, 2019, doi:10.3390/ma12193175_

Round 1
Reviewer 1 Report
The article presents the results of research on the synergistic effect of pH and dissolved oxygen on the corrosion process of X80 steel in a simulated acidic soil solution (Acidic Soil Simulated Solution). The presented test results have an utilitarian character when assessing the corrosive wear of pipelines operated in acid soil.
The introduction well characterized the current state of knowledge with reference to literature and the legitimacy of the research undertaken. Although, I think the last paragraph (lines 58-67) more closely matches Chapter 2 Experimental.
Materials and research methodology are described in detail, and the results of the research are clearly presented with references to the literature from the last few years.
Specific comments to the article:
In line 52 the abbreviation DO appears, which should be explained
Line 113-117, Figure 2
The authors clearly state in the text that they present the results of their previous research and refer to literature (line 115), but I believe that the description under figure 2 should also contain a reference to literature.
Line 171, 174, 175.
Error in drawing numbering (Fig3? Is 2?)
Line 239
50%24 ?
Line 257
Fig. 3d should be checked.
Reviewer 2 Report
Reviewers' comments:
Manuscript ID: Materials-604344
Full Title: Coupling Effects of pH and Dissolved Oxygen on the Corrosion Behavior and Mechanism of X80 Steel in Acidic Soil Simulated Solution.
Comments:
The manuscript describes the Coupling Effects of pH and Dissolved Oxygen on the Corrosion Behavior and Mechanism of X80 Steel in Acidic Soil Simulated Solution. The manuscript needs a detailed editing. Some markings are made to just illustrate the extent of editing needed. A thorough revision addressing all the concerns is needed and if the authors are prepared to do that it can be considered for a review of the revised manuscript.
Use either abbreviation or expansion of oxygen consuming and hydrogen evolution uniformly. Use one format either oxygen-consuming or oxygen consuming. Line numbers: 78, 90, and 239. Why reference numbers are in superscript? Follow journal guidelines for uniform reference format. Check front size of 4 given in 4e in equation 5. Where are the details about publication in 249 and 250. References should be given for section 2.2, and Figure 4 and 5 in section 3.2. Clear labeling of RE, CE and WE should be made for Figure 1a. Line number: 124-126, what does decrease of solution in these lines mean? In Figure 4 - Is that Fe2CO3 or FeCO3? Refer Figure 4 and discuss. Section 3.3 title can be shorten. Please provides the references for all equations and formula. Conclusion- the authors need to improve with more specific short results. English needs revision.
So that I recommended this manuscript to major revision and for future process.
Round 2
Reviewer 2 Report
Reviewers' comments:
Manuscript ID: Materials-604344
Full Title: Coupling Effects of pH and Dissolved Oxygen on the Corrosion Behavior and Mechanism of X80 Steel in Acidic Soil Simulated Solution.
Comments:
The manuscript describes the Coupling Effects of pH and Dissolved Oxygen on the Corrosion Behavior and Mechanism of X80 Steel in Acidic Soil Simulated Solution. The authors revised the manuscript according to the reviewers' comments. So that I recommended this manuscript accept for publication in Materials.